# The Biological Function of Extracellular Vesicles during Fertilization, Early Embryo—Maternal Crosstalk and Their Involvement in Reproduction: Review and Overview

**DOI:** 10.3390/biom10111510

**Published:** 2020-11-04

**Authors:** Emanuele Capra, Anna Lange-Consiglio

**Affiliations:** 1Istituto di Biologia e Biotecnologia Agraria, Consiglio Nazionale delle Ricerche IBBA CNR, 26900 Lodi, Italy; emanuele.capra@ibba.cnr.it; 2Dipartimento di Medicina Veterinaria, Università degli Studi di Milano, 26900 Lodi, Italy; 3Centro Clinico-Veterinario e Zootecnico-Sperimentale di Ateneo, Università degli Studi di Milano, 26900 Lodi, Italy

**Keywords:** extracellular vesicles, exosomes, microvesicles, miRNA, protein, reproduction, embryo, biomarker, diagnosis, therapy

## Abstract

Secretory extracellular vesicles (EVs) are membrane-enclosed microparticles that mediate cell to cell communication in proximity to, or distant from, the cell of origin. Cells release a heterogeneous spectrum of EVs depending on their physiologic and metabolic state. Extracellular vesicles are generally classified as either exosomes or microvesicles depending on their size and biogenesis. Extracellular vesicles mediate temporal and spatial interaction during many events in sexual reproduction and supporting embryo-maternal dialogue. Although many omic technologies provide detailed understanding of the molecular cargo of EVs, the difficulty in obtaining populations of homogeneous EVs makes difficult to interpret the molecular profile of the molecules derived from a miscellaneous EV population. Notwithstanding, molecular characterization of EVs isolated in physiological and pathological conditions may increase our understanding of reproductive and obstetric diseases and assist the search for potential non-invasive biomarkers. Moreover, a more precise vision of the cocktail of biomolecules inside the EVs mediating communication between the embryo and mother could provide new insights to optimize the therapeutic action and safety of EV use.

## 1. Introduction

In mammals, the female reproductive trait, comprising ovaries, oviducts and uterus, plays a crucial role in the regulation of early and late reproductive events and provides the optimal environment for embryonic development. Secretory extracellular vesicles (EVs) are important for cell to cell communication, as they have a paracrine function being able to diffuse over a relatively short distance and produce a local action. It has been largely demonstrated that EV-associated activity is fundamental to reproductive success, mediating the fine-tuning of cellular cross-talk in the reproductive system, promoting embryo implantation and assisting successful pregnancy [1]. Especially in reproduction fields, many early studies used low-specific EV isolation methods that often co-isolated other soluble molecules such as growth factors, cytokines and metabolites, making difficult to attribute the activity specifically to EVs and not instead to low amount of other highly active soluble mediators. Extracellular vesicles (EVs) are membrane bound organelles which can convey information between cells through the transfer of functional protein and genetic information to alter phenotype and function of recipient cells [2,3]. Many recent reviews highlight the role of EVs in oogenesis, oocyte maturation and fertilization, embryo-maternal cross talk in the oviduct and embryo implantation [1,4,5,6,7,8,9,10,11,12,13]. In this review, the recent knowledge on maternal-fetal communication, with special reference to the role of EVs during early interaction between embryonic and maternal tissues are reviewed. At first, we summarize the methods used for EV isolation and their molecular and functional characterization to underline the complexity of multi-signaling response in EV-mediated cell to cell communication. Then, we examine the key role of EVs in early reproductive events and embryo-maternal cross talk, focusing on the molecular cargo of the EVs that transmits messages to different compartments of the reproductive tract and are involved in embryo-maternal cross-talk. Finally, we analyze EVs and their cargo for potential novel diagnostic and therapeutic applications in animal and human reproductive medicine (Figure 1).

## 2. Extracellular Vesicles and Biogenesis

Membrane-enclosed microparticles were first isolated from biological materials such as animal platelet-free plasma [14], human seminal plasma [15] and from a variety of mammalian cell lines [16]. The isolated vesicles were assumed to originate from outward budding of plasma membranes. It was not until years later that a novel pathway involving active vesicle secretion was described in reticulocytes and the term exosomes was used to define intracellular endosome vesicles released by exocytosis [17]. Although EVs have been isolated from all types of organisms including Archaea [18], Bacteria [19] and Eukarya [20], in animals they were classified into two main categories: exosomes and microvesicles depending on their size and biogenesis. The smallest exosomes (ranging in size between 50 and 150 nm in diameter) originate from the multivesicular endolytic compartment by the fusion of multivesicular endosome with the plasma membrane [21,22]. Microvesicles (ranging in size between 50 nm and 1,000 nm) are shed by an active outward budding mechanism mediated by reorganization of the actin-myosin cytoskeleton [23,24]. 

## 3. Methods for EV Isolation and Characterization

Cells produce and release a heterogeneous spectrum of EVs differing in size and chemical and physical characteristics. Many methods based on centrifugation, filtration, precipitation and affinity interaction have been used to isolate EVs [25]. The isolation of EVs from complex body fluids such as blood or follicular fluid often requires a combination of different techniques such as differential centrifugation, density gradient centrifugation, filtration, affinity-based method precipitation with polymers such as polyethylene glycol and size exclusion chromatography [25,26,27,28,29,30,31]. Recently, a new type of small (<50 nm), non-membranous nanoparticle named exomeres was separated from various secreted vesicles using asymmetric flow field-flow fractionation technology and ultracentrifugation [32,33]. Isolation efficiency is dependent on the type of sample and the research purposes and procedures must be carefully standardized [34].

Each method present advantages and disadvantages and should be carefully chosen based on sample type, downstream application and scientific question. Some indications are reported in the guidelines presented by international EV body, ISEV [34,35] and summarized in Table 1. 

All procedures include methods for evaluation of vesicle morphology, size distribution and concentration so that the purified EVs can be correctly identified. A topographical EV view is obtained by direct imaging methods including scanning (SEM),transmission electronic microscopy (TEM) and cryo-electron microscopy (cryo-EM) that provide high resolution imaging and a more precise evaluation of vesicular size [36,37]. However, both techniques require laborious sample preparation procedures that limit sample throughput. Methods based on measurement of the Brownian motion of suspended particles include nanoparticle tracking analysis (NTA) and dynamic light scattering (DLS), or flow cytometry (FC) and tunable resistive pulse sensing (TRPS) methods can be used to measure EV concentration in solution or for determination of the EV size ranges when high-throughput information is desired. However, the detection of EVs becomes difficult when large vesicles are present in dynamic light scattering methods [38,39]. Noble et al. [40] reviewed a comparison between electron microscopy and optical methods for EV detection. Isolated vesicles are likely to be a combination of microvesicles and exosomes varying in size [40]. Thus, further biochemical characterization is needed to provide information on EV composition. Extracellular vesicle preparation identity and purity must be evaluated by detecting the presence of anchored protein localized at the external membrane such as transmembrane (CD63, CD81, CD82, CD47) or GPI-anchored protein (CD73) and the absence of major constituent proteins of non EV structures that co-isolate with EVs such as albumin (ALB) and apolipoproteins A1/2 and B (APO1/2, APOB). In addition, to evaluate EV integrity of lipid bilayers the presence of cytosolic protein (ALIX, heat shock proteins HSPs) should be taken into consideration [35]. Subtype characterization can be obtained by detecting the presence of proteins associated to different intracellular compartments. Extracellular vesicles subtypes separation from peripheral blood confirmed the specific isolation of microvescicles using specific antibody targeting proteins, reflecting their biogenesis such as tubulin, actinin-4 or mitofilin for microvesicles and tetraspanin antibodies (e.g., CD9, CD81) for exosomes [41].Protein markers for EV characterization are detected using different technologies. Western blotting and enzyme-linked immunosorbent assay (ELISA) are the most commonly used method to evaluate the presence of EV associated markers. In general, both techniques require a large sample volume and long processing. The limit of detection is similar for both methods, but ELISA can be easily scaled up for higher-throughput measurements. In alternative, EV protein composition can be assessed by mass spectrometry, that provides quantitative and comparative EV proteomic characterization, although requires significant preparatory and processing time. Surface protein marker can be detected using other methods such as small particle flow cytometry, surface plasma resonance. Methods for EV protein characterization were extensively reviewed by Shao et al. [42].

Extracellular vesicles represent an important mode of intercellular communication by facilitating the horizontal cell to cell transfer of lipids, proteins, RNA species and DNA fragments. However, the functionality and characterization of EV molecular cargo is not always easy to interpret due to the presence of different isolate subtypes and the inability to further separate and to stratify these into distinct sub-populations [43,44].

## 4. Extracellular Vesicle Composition and their Molecular Cargo Function

Lipidomics and proteomics have identified about 2000 lipid and 3500 protein species isolated from EVs secreted by different cell sources [45,46]. Extracellular vesicle lipid composition is specific for cellular type. Extracellular vesicles can transfer lipid mediators that regulate several pathologic processes including inflammation, tumor development, and atherogenesis [47]. Extracellular vesicles contain a subset of cellular proteins derived from plasma membranes, endosomes, and cytosol that are consistent with their biogenesis and cellular type. Among them, specific proteins can be used to distinguish between different EV classes and subtypes [48]. Extracellular vesicle surface proteins, such as glycan binding proteins, are probably used by the vesicle for cell specific targeting. The presence of amyloid-β (A4) protein outside EVs mediates the destination of EVs from neuroblastoma cell to neurons in the central nervous system. At the same time, the expression of some proteins such as beta amyloid peptide, synuclein and TAR DNA-binding protein 43 (TDP43) changes in neuronal disease and can be used as a marker for progression [49]. Extracellular vesicle proteins can also mediate cellular communication. Surface proteins present on EVs released by the inner cell mass such as laminin and fibronectin can interact with integrins on the surfaces of the trophoblast promoting embryo implantation [50]. 

The emerging role of cell to cell communication is mediated by EV-associated small non-coding regulatory RNAs called microRNAs (miRNAs) and other non-coding RNAs by acting as “messengers” shuttled between different cell types. In fact, miRNAs can be specifically compartmentalized inside vesicles and released outside cells, as previously reported in mesenchymal stromal cells in human and equine studies [51,52]. However, nonselective miRNA loading and secretion has also been reported [53,54]. The relative contributions of passive and active packaging of RNAs into EVs is not fully understood [55]. In addition, stoichiometric analysis of EV miRNA cargo indicate that EVs isolated from different tissues do not contain many copies of miRNAs [56]. This unexpected result may be explained by the fact that isolated EVs represent a heterogeneous population of vesicles with different physical and molecular properties [57,58]. 

Many aspects of how miRNAs are selectively encapsulated in EVs and can mediate cell to cell communication are still not understood [59]. Molecular profiling of EVs surface revealed specific molecular composition that mediate interactions at the surface of extracellular vesicles [60,61]. It is likely that EVs recognize and transfer their molecular content such as miRNAs proteins and lipids, through an active internalization process mediated by various endocytic mechanisms such as clathrin dependent and caveolae dependent endocytosis and lipid raft mediated uptake. In fact, antibodies against specific proteins such as tetraspanins, integrins, immunoglobulins, proteoglycans, lectins block vesicular entry into recipient cells [62,63].

Other nucleic acids, including mRNAs and DNAs, are important in regulation of gene expression but their role in the transfer of cell to cell information remains controversial. Transcripts have been observed to be delivered by EVs but rapidly degraded, whereas plasmid DNA can only be delivered by EVs for correct functioning in recipient cells [64]. Recently, EVs were observed to mediate horizontal gene transfer during genome editing. The use of the CRISPR-Cas9 system in NIH-3T3 mouse cells showed an unexpected insertion of bovine DNA, derived from satellite DNA elements, that was mediated by EVs present in the cell culture medium [65]. Double-stranded DNA in EVs has also been used in medical diagnostics as a biomarker for cancer [66,67].

## 5. Extracellular Vesicle in Animal Reproduction and Embryo-Maternal Cross-Talk

Cell communication is fundamental for several molecular processes that allow populations of cells to exchange information with one another to aid specialization in tissue or coordinate multiple cross-organ interaction. 

Extracellular vesicles have been identified as one of the key players in regulating temporal sequences and spatial interaction, cell-to cell signaling in all events in sexual reproduction (i.e., gametogenesis, fertilization and embryogenesis) and in embryo-maternal cross-talk (Figure 2 and Table 2). 

Extracellular vesicles in seminal fluids were observed to modulate sperm capacitation in man [82] and pigs [83] and to influence female physiology by modulating immune-related gene expression in the porcine endometrium [68]. In return, oviductal EV (oEV) secretion was observed to regulate sperm motility and capacitation in mouse [69,70], cat [71] and bovine [72] sperm and EVs from avian uterine fluid may play an essential role in the preservation of sperm functions [81]. During oogenesis, follicle growth and oocyte maturation required a constant signal exchange in which there is a mutual recognition of diffusible or EV-mediated molecules produced by somatic cells (granulosa, cumulus and theca) and by germ cells (oocytes) present in the follicular fluids (FF). Extracellular vesicles isolated from FF were first observed and characterized in horse [84], human [85] and bovine [86] studies. Follicular fluid comprises a heterogeneous EV population secreted by granulosa, cumulus and somatic follicular cells with functions related to control of steroidogenesis [85,87]. Extracellular vesicles from FF were able to modulate bovine embryo development in vitro [75].

The successful of pregnancy requires a molecular dialogue between the embryo and the female reproductive tract that starts at the oviduct and continues until the formation of placenta. The first known molecules in this paracrine communication were the cytokines and growth factors, such as interleukin-1ß (IL-1ß), heparin-binding epidermal growth factor (HB-EGF), integrins and leukemia inhibitory factor (LIF) that act synergistically in the embryo-maternal crosstalk. The IL-1 increases the expression of molecules of adhesion of epithelial cells regulating the endometrial receptivity [88] and stimulating angiogenesis to promote the embryonic growth [89]. The HB-EGF is one of the members of epidermal growth factor family and, through the HB-EGF receptors on the surface of embryo and endometrium, facilitates the process of implantation and promotes the development of blastocyst [90,91]. The expression levels of IL-1β and HB-EGF are higher in implantation phase and are positively correlated with the levels of estradiol and progesterone [92], showing their important role in this phase. Integrins are adhesion molecules and constitute a system of cell-cell and cell-matrix interaction. During implantation, the integrins α6β1 and α7ß1 of blastocyst bind to laminin of the basement membrane of endometrium, while the integrin α4β1 of endometrium interferes with trophoblastic fibronectin. In this way, this new set of integrins promotes the interaction between the trophectoderm and the endometrium and then the implantation [93]. The LIF has different role in embryo implantation and it is produced by both the blastocyst and the endometrium: increases the expression of EGF and implantation genes in receptive endometrium; regulates growth and development of the embryo; increases the production of cytokines and prostaglandins; stimulates trophoblast cell differentiation and increases trophoblast capability to invade the uterine stroma and it is involved in recruitment of specific cohort of leucocytes which participates in uterine inflammatory response during implantation. The action of LIF is due, for example, to the regulation of expression of adhesion molecules such as L-selectins, E-cadherins and tight junction proteins or to the activation of signal transducer and activator of transcription 3 (STAT3) phosphorylation, or to production of prostaglandin E (PGE2) stimulating COX-2 and microsomal PGE synthase-1 (mPGES-1) enzymes expression that are involved in PGE synthesis. Due to its important role, this factor is proposed as non-hormonal contraceptive [94]. The LIF, with other cytokines, chemokines, and chemokine receptors, is present in EVs too [95] but while the soluble secretome is involved on mouse embryo growth, the EVs are implicated on implantation and embryo development and may be implicated in the pathophysiology of implantation failure in infertility [7].

Indeed, the EVs mediate two-way trafficking of molecules for embryo-maternal communication. During embryo migration from the oviduct to the uterus, the embryo undergoes distinct metabolic stages (cleavage, morula, blastula, gastrula). Labelled in vivo EV preparations from bovine oviduct flushing were internalized by in vitro produced embryos [73]. Both in vivo oEVs and in vitro EVs isolated from bovine oviduct epithelial cell culture were shown to exert a positive effect on development and quality of in vitro produced cattle embryos [73,74]. Although a reciprocal regulation mediated by EVs from the embryo has not yet been documented, vesicles from the embryo can regulate embryo development and implantation. In mice, laminin and fibronectin present in EVs derived from the preimplantation blastocyst inner cell mass (ICM), that gives rise to a whole embryo, interact with integrins on the cell surface of trophectoderm (TE) cells, increasing efficiency of blastocyst implantation [50]. Extracellular vesicles exchange in the uterine environment was reported in vivo whereas fluorescent labelled EVs isolated from sheep uterine fluid were internalized in the conceptus trophectoderm and uterine epithelia [96]. During implantation, a synergistic endometrial-blastocyst crosstalk in the maternal uterine environment is necessary for maternal recognition and successful pregnancy [1]. Intrauterine bovine EVs are required for conceptus implantation and can regulate blastocyst development [79,80]. In human, endometrial-derived EVs were observed to be internalized in HTR8 trophoblast cells and in spheroid model of trophectoderm and enhance cell adhesion capabilities [10,11]. In response, trophoblast cells form the interface with the maternal cells and the fetus and can secrete EVs to enable the fetus to interact with uterine endothelial cells. Conceptus-derived EVs containing Interferon tau (IFN-t) isolated from pregnant ewes were able to modulate expression of genes in bovine endometrial epithelial cells (EECs) [77]. Extracellular vesicles derived from porcine trophectoderm-cells induce aortic endothelial cell proliferation providing evidence of their ability to stimulate angiogenesis [76]. Finally, the placenta secretes EVs carrying proteins involved in immune-modulation which is essential for success of pregnancy [97,98]. In vitro, human placental cytotrophoblast cells are observed to respond to hypoxia by modifying the bioactivity of EVs, thereby, promoting the migration of extravillous trophoblasts into the decidua and myometrium and establishing placental perfusion [78]. Placenta-derived EVs are known to contain different pro-inflammatory molecules that potentially can modulate the local endometrial immune system and mediate local immunotolerance of the fetus to evade the maternal immunosurveillance [99]. In fact, cytotrophoblasts from villous release EVs that are observed to reduce the inflammatory response in peripheral blood mononuclear cells (PBMCs) [100]. Interestingly, a concurrent variation of pro-inflammatory state and dysregulation of placenta-derived EVs and total EVs is observed in some complications of pregnancy such as preeclampsia [101]. To note, EVs isolated from preeclampsia patients also inhibit angiogenesis of human endothelial cells [102], suggesting an active role in cell to cell communication for the maintenance of the correct physiological state during pregnancies. 

## 6. Extracellular Vesicle Molecular Cargo in Animal Reproduction and Embryo-Maternal Cross-Talk

The biological function of EVs in regulating early embryo-maternal interaction results from the transfer of critical molecular cargos to distant or neighboring recipient cells or tissues. Although the latest high-throughput analytical platforms can perform detailed analysis of the lipid, glycolic, proteic and nucleic acid content in EVs [103], it is still difficult to establish the molecular basis of EV cargo regulating embryo-maternal communication because of the many variables associated with EV isolation and characterization. Mass spectrometry analysis shows significant differences in the protein content of in vitro and in vivo EVs secreted from the oviduct [73], and between the protein cargo of placental EVs isolated using different procedures [104]. Isolation of different-sized fractions of EVs from human first-trimester placentae, by differential ultracentrifugation, showed that although many proteins involved in vesicle transport and inflammation are shared between different fractions, some proteins related to vesicle internalization, membrane cofactor and minor histocompatibility antigens, are specific for different-sized EVs. Therefore, it is likely that different sized EVs can interact with different maternal cells mediating multiple messages during feto-maternal communication in early pregnancy [105]. In addition, the overall EV molecular cargo in bovine oviductal fluid changes in function in different stages of the estrous cycle, suggesting that EV composition is under hormonal control [106]. A recent review by Almiñana and Bauersachs, which compared the protein and miRNA content in oEVs, based on literature available data, found only a partial overlap of EV molecular cargo between different species probably because of different EV isolation and characterization methods [5]. 

Several proteins in EVs regulate different reproductive functions promoting fertilization and embryo implantation and growth. Extracellular vesicles released into the murine uterine luminal fluid deliver sperm adhesion molecule 1 (SPAM1) to sperm membranes, enhancing sperm fertility [107]. Extracellular vesicles from the oviduct transfer plasma membrane Ca2+ ATPase 4 (PMCA4) protein to sperm to enhance Ca2+ efflux in mice [83]. Embryo EVs transport progesterone-induced-blocking factor 1 PIBF that increases IL-10 production in maternal lymphocytes, sustaining immune responses during pregnancy [108]. In return, placental tissue from the first trimester releases EVs containing proteins with roles in regulating and modulating T-cell activity to modify the maternal immunological environment, such as Fas ligand, TNF-related apoptosis-inducing ligand TRAIL, programmed death-ligand 1PD-L1, B7 homolog 1 B7-H1, B7 homolog 3 B7-H3 and human leukocyte antigen G 5 HLA-G5 [109]. 

MicroRNAs encapsulated in secreted EVs play a significant role in intercellular communication reaching target cells and regulating mRNA and protein expression. Extracellular vesicles transferred from the embryo to the mother, or vice versa, transport miRNAs that target specific pathways for fertilization, embryo implantation and fetal development. Following hormone stimulation, in silico bioinformatic analysis showed that miRNA content, which potentially regulates critically important pathways for follicular oocyte maturation, changed in follicular fluid EVs [110]. Extracellular vesicles isolated from bovine follicular fluid contain miRNAs that reflect the stage of the estrus cycle and can modulate cumulus cell transcription during in vitro maturation [111]. Murine oEVs can transfer miR-34c-5p to the sperm heads where it localizes to the centromere and promotes the first cleavage in the zygote [112]. In addition, oEVs carry miRNAs that potentially target several embryonic development related genes [112]. Oviductal EVs were observed to fuse with canine oocytes and release miRNAs with a potential role in oocyte maturation and follicular growth [113]. In vitro, oEV supplementation altered the bovine transcriptome, suggesting a possible role of oEV miRNA cargo in controlling embryonic development [114]. Extracellular vesicles secreted by donor oviductal cells increased birth rates after embryo transfer in mice due to decreased apoptosis and improved cellular differentiation in embryos [115]. A miRNA mediated embryo-maternal communication was also reported and Cuman et al. (2015) [116], demonstrated that human embryos secrete miR-661 that was up-taken by endometrial epithelial cells and inhibited cell adhesiveness [116]. To note, miR-661 exchange was observed to be mediated by the RNA binding complex (RBC) protein Argonaute 1 rather than EV shuttling [99], but it is likely that other EV encapsulated miRNAs can potentially be shuttled from embryos to maternal tissue. Recently, the presence of microRNAs and extracellular vesicles in human blastocoel fluid was reported, suggesting a possible role of EV miRNA cargo in regulation of blastocyst development. MiRNA expression in blastocoel fluid reflects the miRNome of embryonic cells with abundant miRNA targeting genes regulating critical signaling pathways controlling embryo development [117]. Syncytial-like cells, derived from murine trophoblast stem cells, secrete EVs with a miRNA transcript from the X chromosome cluster miR-322-3p and 5p, miR-503-5p, miR-542-3p, miR-450a-5p, miR-292-3p and mmu-miR-183-5p that was found to be increased in EVs isolated from the sera of pregnant mice [118]. In vitro and in vivo studies have reported miRNA EVs shuttling from endometrium to embryos. During the window of implantation, human endometrium epithelium releases EVs carrying specific miRNAs [8]. Among them, miR-30d was observed to be encapsulated in EVs and released by primary human endometrial epithelial cells (hEECs) and transferred to the trophectoderm of murine embryos when co-cultured, thus regulating genes related to embryonic adhesion and promoting embryo implantation [8]. MicroR-34c-5p was found to be differentially enriched in EVs isolated from mouse endometrium during different phases of pregnancy, and it is likely to target GAS1 (Growth Arrest Specific 1) and influence embryo implantation [119]. In sheep, a clear influence of progesterone-mediated regulation of miRNA content in EVs isolated from the uterine lumen was reported [120]. An RNA-based communication occurred between uterus and embryo also through functional mRNA-encapsulating EVs. In vitro study of co-culture of human trophoblast spheroids and human endometrial cell lines showed that EVs package and transport specific RNAs (intronic and exonic LINC00478, and ZNF81) from trophoblast to endometrial cells, that results in a reduced level of the corresponding transcripts in the endometrial receiving cell [121]). Finally, Simon et al. [122] first characterized the DNA-cargo of the EVs by the embryo showing a random representation of embryo genomic DNA inconsistent with aneuploidy or apoptotic cellular events [122].

## 7. Extracellular Vesicles as Biomarkers in Reproductive Medicine

The EVs are important vectors of information and are involved both in physiological and pathological conditions. Indeed, during diseases, cell metabolism, physiology and behavior are modified causing alteration in cellular gene expression, protein content and EV cargo. This makes EVs important in the scientific field for their functional role in cell communication and in the medical field as biomarker or as therapeutic tools.

The physiological level of circulating EVs in tissues, serum or other biological fluids is different considering the animal model, the time (hours or days) after disease, the kind of disease and the EV quantification method. However, this information, studied and identified in the different conditions, could help to identify the possible therapeutic dose.

In the reproductive tract, EV-mediated communication between cells and embryo–maternal crosstalk is an essential process that occurs both in physiologic and in pathologic conditions. Therefore, molecular cargo of EVs and its release change in the pathologic condition by affecting maintenance, and progression of reproductive- and obstetric-related pathologies (Table 3).

### 7.1. Biomarkers for Female Reproductive Cancer

Extracellular vesicles can regulate a variety of physiologic processes. The molecular cargo of EVs is strictly dependent on the status of the cell or tissue donor and can change in pathologic conditions such as cardiac disease, neurologic diseases and cancer [132]. Although, tissue biopsies are the gold standard for cancer evaluation, they are invasive and provide only a small sample of the whole tumour. Moreover, it can be difficult to collect sequential samples to monitor the progression of the tumour with time and response to therapy [133]. 

Extracellular vesicles eliminate these limitations. They are present in all biological fluids (including urine, blood, saliva and cerebral spinal fluid), and their involvement in different diseases make them inviting candidates as biomarkers. Collection of body fluids for screening of some diseases could avoid the necessity for invasive examinations [134]. Inside the EVs, the molecular cargo is protected from degradation and can be used as biomarkers for non-invasive cancer diagnosis [135,136]. For example, specific extracellular vesicle-derived miRNAs are associated with eight types of cancer (including lung, breast, and ovarian) [137]. In addition, analysis of miRNA from some EVs may be used to distinguish between benign and malignant disease. Indeed, expression levels of miR-200 family members (miR-200a, miR-200b and miR-200c) in one study could distinguish malignant from benign ovarian tumours with a sensitivity of 88% and specificity of 90% [123]. The strength of miRNAs as ovarian tumour markers is confirmed by the fact that elevated miR-200b and miR-200c levels are correlated with serum CA125 levels which is a current standard of diagnosis, monitoring treatment reaction and predicting prognosis [123]. 

### 7.2. Biomarkers for Female Fertility

Physiologic alterations occurring in the reproductive tract in pathologic conditions influence cell metabolism, changing gene expression and protein content and thus altering EV concentration, cargo and function. These alterations influence maintenance and progression of reproductive- and obstetric-related pathologies.

In human medicine, decrease in fertility is an important issue. This may be correlated to environmental stressors. Phthalates and phenols are classes of potential endocrine disrupting chemicals present mainly in the environment and in food [138]. Phthalates and phenols have been shown to influence fertility in women [139]. A correlation has been identified between the concentration of phthalates and phenols and their impact on the EV-miRNA profile in follicular fluid with a negative impact on female fertility [124]. 

Several of the significant EV-miRNAs biomarkers found to be associated with phthalates and phenols play key roles in the ovarian follicle, consistent with the fact that the follicular fluid, contains a number of molecules including EV-miRNAs that create an appropriate microenvironment for the growth of oocytes [140,141]. For example, increased urinary concentration of phenols has been associated with altered expression of miR-125b, miR-24 and miR-375, which have been shown to play an important role in animal oocyte maturation and fertilization. In rodents, over expression of a miR-125b mimic block expression of specific genes required for embryos to progress beyond the two-cell stage [142]. Higher levels of miR-24 in bovine culture media is associated with embryos failing to undergo differentiation [143]. MicroR-375 is expressed in granulosa cells and oocytes and targets genes that regulate follicular growth proliferation, spread and apoptosis of cumulus cells [132,133]. Overexpression of miR-375 blocked the ability to proliferate, increased the apoptosis rate of cumulus cells in cows [144,145] and suppressed estradiol production and follicular development in porcine granulosa cells [146]. 

These data show that exposure to toxic compounds causes miRNAs alteration in human EVs isolated from follicular fluid, potentially affecting ovarian function. In the veterinary field, it is likely that exposure to toxic compounds such as Fusarium mycotoxins, frequently present in feed, might influence animal fertility and alter the miRNA expression in follicular fluid EVs. 

In man, but probably also in animals, fertility status is influenced by other factors such as advanced maternal age (AMA) or endometriosis that can impact the embryo-endometrial talk before implantation. In vitro studies revealed variation of EV-bound secreted miRNAs isolated from co-cultured endometrial cells and embryos from AMA and fertile controls, with 16 miRNAs increase in AMA and strong evidence of a negative regulation toward 206 target genes involved in cell adhesion. Co-cultured endometrial cells isolated from patients with endometriosis and embryos from fertile controls revealed EV alterations in 10 miRNAs, targeting 1014 genes crucial for development and implantation [125].

### 7.3. Biomarkers for Embryo Quality

Currently, in vitro embryo production is an important component of improved pregnancy rates in assisted reproduction in both human and veterinary medicine. Blastocyst quality is assessed morphologically by assessment of the expansion of the blastocoel cavity, as well as the appearance of the ICM and TE cells [147,148,149,150]. Another important technique is removal of cells from the TE by biopsy [151]. In human medicine, this method is used to detect, and thus prevent, transmission of single gene disorders such as cystic fibrosis and β-thalassemia [152]. The identification of specific markers for embryo quality, would avoid the need for embryo biopsy and its inherent risks. In 2013, for the first time, genomic DNA of unknown origin was identified in blastocoel fluid (BF) and in embryo culture medium [153]. However, the use of mitochondrial DNA to assess embryo quality is not used because of the presence of other contaminant DNA present in the culture media from maternal (cumulus) cells [154].

The miRNAs found outside the blastocyst in the spent medium are probably involved in implantation and in embryo-maternal cross-talking [116,155,156,157]. Since embryos secrete miRNAs in culture medium in vitro and in the uterus in vivo, it is conceivable that the extracellular vesicles, carrying miRNAs, DNA and other molecules, might reflect embryo quality [85]. Indeed, some authors show that the expression profile of miRNAs reflects embryo aneuploidies and that the miRNAs can be used to evaluate embryo implantation potential [140]. Extracellular vesicle cargo released in spent culture medium depends on the physiologic state of the embryo and reflects the in vitro embryo quality before implantation or its cultural environment. For example, during in vitro embryo culture, EV concentration in low oxygen tension increased on Day 3 and decreased on Day 7 and miR-210 levels can be used as a marker for normoxia since it is associated with low oxygen tension [158].A method (based on characterizing EVs by flow cytometry), using propidium iodide for DNA staining, showed that low numbers of stained vesicles is associated with embryo competence indicating reduced cell injury and embryo damage and thus less degraded DNA in the EVs [126]. MicroRNA profiling of spent blastocyst culture media from implanted versus unplanted blastocysts showed increased expression of specific miRNA in degenerate conditioned media in bovine [138] samples, probably associated with different EV cargo. However, Cimadomo et al. [159] reported that medium from poor-quality blastocysts contained more miRNAs probably as a result of passive miRNA accumulation in the medium because of cellular turnover rather than active secretion from EVs.

### 7.4. Biomarkers for Placenta Quality

Pregnancy is a particular physiological condition for studying EVs. At first, unlike other conditions, the onset and end of pregnancy is known allowing you to follow changes in EVs throughout the period. In addition, the placenta produces specific markers that allow you to distinguish EVs from those produced by others cell types and, finally, the placenta is available at the end of pregnancy for study [160]. Placental secreted EVs are important in feto-maternal communication and have been identified in maternal blood. Placental EVs were isolated from maternal blood by a chromatographic/immunosorbent procedure using antibody agarose beads against placental alkaline phosphatase (PLAP), a protein present exclusively on EVs secreted from placentae [161]. Placenta-derived EVs may be differentiated from other EVs by the presence of placenta specific proteins (eg, PLAP4) and miRNAs (eg, chromosome 19 miRNA cluster) that are exclusively expressed in the placenta [53,161,162].

During gestation, placenta-derived EVs can be bio-markers of feto–maternal health and evolution, indeed, EVs are involved in cell-to-cell communication between the placenta and peripheral blood immune cells. During the first trimester of pregnancy, extravillous trophoblast and/or syncytiotrophoblast cells release placental EVs into the maternal blood. These EVs suppress maternal T-cell signaling components promoting immunosuppression and maternal immune tolerance to the fetus. In man, the plasma concentration of EVs increases during pregnancy and with gestational age, indeed, the concentration of EVs in maternal peripheral blood is 20-fold greater compared to that found in nonpregnant women [125]. The increased level of EVs is also found in several pathologic conditions [163] and for this reason changes in exosome levels may be of clinical utility in the diagnosis of placental dysfunction. Indeed, it has been demonstrated that some proteins (for example the specific syncytiotrophoblast protein, syncytin-2) are markedly down-regulated in EVs derived from the placentae of pregnant women with preeclampsia compared to healthy control pregnancies [127]. For this reason, placental EVs and specific miRNAs have been further examined as biomarkers for diagnosis of pre-eclampsia [128]. Plasma EVs isolated from pre-eclampsia pregnancies were shown to impair angiogenesis of human umbilical vein endothelial cells and to expressed abundant sFlt-1 (soluble fms-like tyrosine kinase-1) and sEng (soluble endoglin) [102]. 

In addition, the secretion of EVs from the placenta can be influenced by the microenvironment. Indeed, variation in oxygen tension and glucose can increase bioactivity and release of exosome by trophoblast [98,164]. Extracellular vesicles isolated from placenta of pregnant women with diabetes mellitus show alterations in the protein content related to energy production, metabolism and inflammation [129] and alteration in miRNA associated with skeletal muscle insulin sensitivity [130].

Sometime, the effects of high glucose concentration on the bioactivity of EVs are additive to those of oxygen tension when there is an association with maternal insulin resistance (resulting in hyperglycemia) and pre-eclampsia (associated with placental insufficiency and hypoxia) [165]. The identification of such biomarkers would be useful screening tests to identify those asymptomatic women who develop common pathologies [165].

### 7.5. Biomarker for Early Abortion

Sabapatha et al. (2006) evaluating the content of miRNAs in pregnant and control women report that the plasma concentration of EVs increases during pregnancy and with advancing gestational age [161]. Pohler et al. (2017) found 194 and 211 circulating extracellular vesicle-derived miRNAs from Days 17 and 24 of gestation confirming their increase during gestation [131]. However, three miRNAs (miR-25, -16a/b, and -3596) were present in greater abundance either in controls or in cows suffering embryonic-mortality compared to pregnant animals on Day 17. The increase in miR-25 abundance in cows with embryonic-mortality might indicate early embryonic death or a systemic response to pregnancy loss. Pathway analysis indicated that the changes to specific miRNAs in cows with embryonic-mortality versus pregnant animals could affect the up-regulation of pathways involved in prostaglandin production [166]. It is known that prostaglandin is responsible for corpus luteum regression and that with this involution the concentrations of progesterone, critical for pregnancy establishment, fall [167,168]. In bovine pregnant female blood, some miRNAs can be down or up-regulates in circulation exosomes demonstrating the potential role of circulating exosomal miRNAs as biomarkers in early embryonic mortality or early pregnancy diagnosis [169]. It would be interesting to validate differentially abundant miRNAs to determine biomarkers for the diagnosis of early pregnancy success or embryonic mortality [131].

## 8. EVs and Therapeutic Action

Since endometrial exosomes are implicated in the pathophysiology of implantation failure in infertility [7], current research is attempting to exploit the clinical utility of EVs. Extracellular vesicles are easy to manipulate because they are composed of biological membranes that protect their contents. Small molecules, proteins or RNA, anti-cancer agents, anti-inflammatory compounds, miRNA, mRNA, proteins, and growth factors can be loaded into EVs via transfection or electroporation [170,171]. EVs can be loaded using endogenous or exogenous approaches. The endogenous method first loads therapeutics into parent cells, followed by the generation and release of loaded EVs [172]. In the exogenous approach, therapeutic agents are incorporated into isolated EVs through an incubation process [173,174]. 

The EVs can be used for treatment of cancer and can be loaded with anti-cancer agents (in cells or in EVs) including doxorubicin, paclitaxel, curcumin, cisplatin, and methotrexate [175]. Extracellular vesicles have had enormous therapeutic potential in neurologic disorders [176], and show regenerative effects in animal models of neurological, cardiovascular, liver, kidney, and skin diseases [177]. In addition, their ability to promote re-epithelialization of skin wounds and stroke recovery has been reported [178]. Recently, amniotic derived EVs were used to treat endometritis in mare with the goal of achieving successful pregnancy [179]. This mare was an 11-year-old Friesian, with a history of failed pregnancies despite numerous insemination attempts for many years. After a final insemination attempt using a stallion of proven fertility, the collection of an eight-day old embryo suggested that the mare was affected by implantation failure related to endometritis. The regenerative medicine using amniotic derived microvesicles was a new approach for this disease. The mare was treated with two cycles of intrauterine administration of amniotic-derived microvesicles that induced an improvement in the classification of endometritis and a pregnancy ended with the birth of the foal. This is the only paper the used EVs in reproductive disease. Probably, amniotic microvesicles for their anti-inflammatory and regenerative effects [52] were able to restore the injured endometrium and re-establish the proper communication for a successful embryo implantation.

Extracellular vesicles can be administered via intravenous, subcutaneous, intraperitoneal, oral, and intranasal routes to reach the desired in vivo target sites, depending on the intended therapeutic action and tissues to be targeted [180]. In the paper by Lange-Consiglio et al. [179] EVs were administered by the intravaginal rout. The biodistribution and targeting potential of EVs is dependent on EV sizes, surface markers, and their isolation methods. Limitations influencing EV biodistribution through various routes of administration include transport through several specific physical barriers, non-target tissue uptake, immune cell-dependent phagocytosis, and rapid in vivo clearance [180].

There are many considerations for optimization of the therapeutic action of EVs and to ensure treatments are safe. These include: 1) the cell type (such as the tissue source of MSCs); 2) the cell collection process and the expansion methods including culture conditions (growth medium and supplement used, type of bioreactor, number of passages performed, etc.); 3) the EV isolation and storage methods; 4) the modality of drug-loading and the quantification approaches; 5) the mechanism used to trigger EV release; 6) the design of cost-effective and scalable isolation processes and 7) the methods to ensure quality, safety, and consistency that are mandatory for clinical applications [180]. Variations in these parameters can influence the EV population size, generation (yield/cell), membrane markers, EV contents including miRNAs, the purity and safety profile of EVs [181] and conceivably their therapeutic action too.

## 9. Conclusions

This review summarizes the role of EVs in cell to cell communication in animal reproduction. Remarkable progress has already been made, but many hurdles remain. Although, there is clearly evidence that EVs play crucial roles in many aspects of oocyte maturation, fertilization, implantation and embryo-maternal crosstalk, challenges remain in a partial understanding of which signaling biomolecules carried by EVs act to mediate cell to cell communication. In fact, cells release a heterogeneous population of EVs differing in size and molecular content that are often co-purified using conventional isolation methods. Molecular profiling gives an overall characterization of EV cargo that only partially represents the function of specific EV subpopulations in physiologic and pathologic conditions. The development of more reliable isolation methods and more sensitive technologies for physical and molecular characterization of EVs is needed to obtain more robust results that will improve our insight into the molecular basis of EV targeting of, and action on, recipient cells. Moreover, a more precise vision of the cocktail of molecules inside EVs that are transferred in the two-way trafficking between embryos and mothers will improve the identification of potential non-invasive biomarkers for specific pathologic conditions and will advance the use of EVs in innovative therapies in reproductive biomedicine.

## Figures and Tables

**Figure 1 biomolecules-10-01510-f001:**
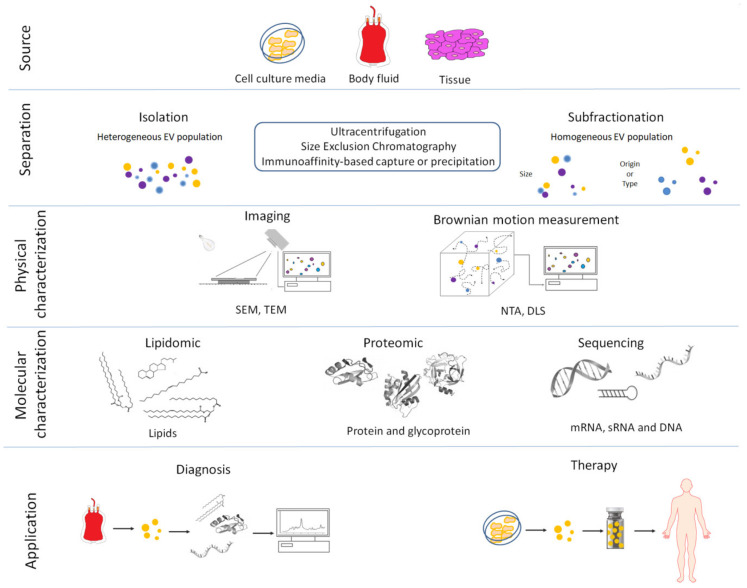
Overview of the main procedures used for the isolation, characterization and clinical use of extracellular vesicles (EVs). (1) Extracellular vesicles are released by cells into the culture media or from tissue into the extracellular environment. (2) Extracellular vesicles are separated or further purified to obtain a more homogeneous EV population using a variety of methods. (3) Isolated EVs are physically characterized by: Scanning Electron Microscopes (SEM), Transmission Electron Microscopes (TEM), Nanoparticle Tracking Analysis (NTA) and Dynamic Light Scattering (DLS). (4) Extracellular vesicles are molecularly characterized using several techniques. (5) Biomarkers can be obtained by molecular profiling of isolated EVs from in vitro cell cultures or body fluids in pathologic condition and used as diagnostic tools for several human and veterinary diseases. Isolated EVs can also be used as treatments in human and veterinary medicine.

**Figure 2 biomolecules-10-01510-f002:**
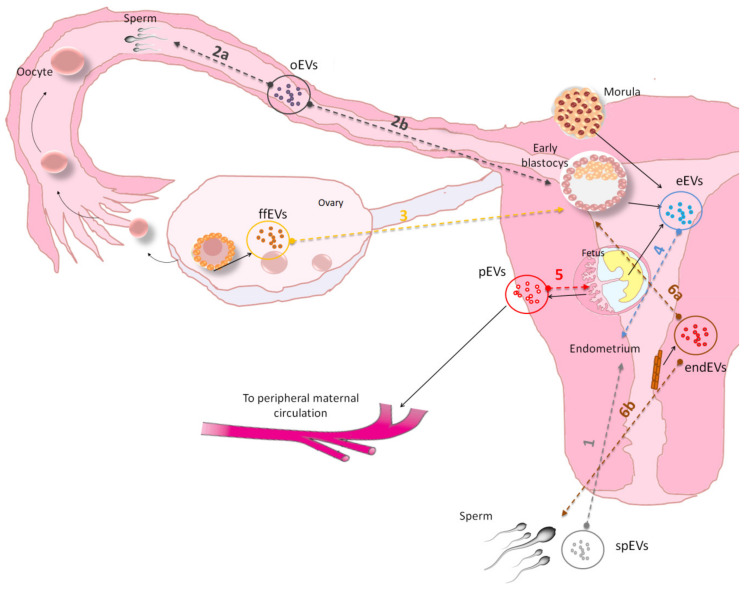
Graphical representation of the EVs mediated cross-talk interaction in the female reproductive system. (1) Seminal plasma EVs (spEVs) interact with endometrium (2) oviductal tract EVs (oEVs)with sperm (2a) and embryo (2b),(3) follicular fluids EVs (ffEVs) with embryo (4) embryo EVs (eEVs) with endometrium (5) placenta EVs (pEVs)with extravillous trophoblasts (6) endometrium EVs (endEVs) with embryo. Colored dashed line (and numbers) indicates EVs release and uptake from donor and recipient cells or tissue, which studies are reported in Table 2. In figure, the human female reproductive tract was shown as example of mammalian female reproductive system.

**Table 1 biomolecules-10-01510-t001:** Summary of EV isolation techniques and main advantages (pros) and disadvantages (cons) for different methods.

Method	Principle	Advantages	Disadvantages
Differential centrifugation	EVs isolation after different consequent centrifugation steps (from 300g to 100,000 g): depletion of cells, platelets and large apoptotic bodies by low-speed centrifugation steps. Larger EVs are pelleted at 10,000 20,000 g range. Smaller EVs are then pelleted at high speed (100,000 120,000 g).	Most used, intermediated recovery and specificity	Time consuming and extravesicular proteins complexes/aggregates, lipoprotein particles, and other contaminants may also sediment
Density gradient centrifugation	EVs isolation through density gradients of sucrose or iohexol or iodixanol	High purity (EVs float upward or move downwards into an overlaid density gradient)	Applicable to small volume (usually after differential centrifugation), sucrose or iohexol or iodixanol can influence downstream application
Filtration	EVs filtration with different molecular weight cutoff	Recovery and purity depend on the consequent centrifugation step and the cutoff of centrifugal filter employ	Low specificity, EV populations may adhere to the filter or filtering may cause deformation and breakup of large vesicles
Precipitation	EVs are precipitated with organic solvent or in presence of different chemical compound such as polyethylene glycol (PEG), sodium acetate or protamine	High recovery, fast and easy	Low specificity Coprecipitation of numerous non-EV contaminants such as lipoproteins. Rigorous assessment of contaminating particle is recommended
Size Exclusion Chromatography	EVs are separated by their ability to pass through a resin packed in a column	Well separated EVs from protein complexes biofluids	Not suitable for large volume
Affinity isolation	EVs bind specific antibodies against exosome-specific surface proteins or EV-binding molecules	High purity	Low recovery, requires specific antibody
Microfluidic devices	Microfluidics-based on-chip EVs isolation based on immunoaffinity, membrane filtration, nanowire-based traps, nano-sized deterministic lateral displacement, viscoelastic flow and acoustic isolation	high-throughput and low processing time	Not suitable for large volume
Nanoscale flow cytometric sorting	Fluorescent labelled EVs are sorted using flow cytometer	Very specific and high purity	Laborious and time consuming

**Table 2 biomolecules-10-01510-t002:** Extracellular vesicles mediated cross-talk between different compartments of the reproductive system as shown in Figure 2. Extracellular vesicles released from different cell or body fluids (seminal plasma, oviduct, follicular fluid, embryo, placental mesenchymal stem cells, endometrial epithelial cells) and from different species (*bos taurus*, *felis catus*, *homo sapiens, mus musculus*, *ovis aries*, *sus scrofa*), target different cell types. Method of isolation, main results and references for each EV-mediated exchange are also shown.

Ref. Figure 2	EVs Isolated from	Species	Isolation Method	Target Cell	Physical Characterization	Main Results	References
1	seminal plasma	*sus scrofa*	polymer precipitation	endometrial epithelial cells	TEM, NTA	induction of immune-related gene expression in endometrial epithelial cells EECs	[68]
2a	oviduct	*mus musculus*	ultracentrifugation	Sperm	TEM	PMCA4 sperm up-take from exosomes released in female luminal fluids	[69]
2a	oviduct	*mus musculus*	ultracentrifugation	sperm	TEM	oEVs transfer increase PMCA1 and PMCA4 activity in sperm	[70]
2a	oviduct	*felis catus*	polymer precipitation	Sperm	NTA	oEVs contain protein related to energy metabolism, sperm functionality and enhance sperm motility and fertility in vitro	[71]
2a	oviduct	*bos taurus*	ultracentrifugation	Sperm	DLS	oEVs induced acrosome reaction and signalingevents associated with sperm capacitation	[72]
2b	oviduct	*bos taurus*	ultracentrifugation	Embryo	TEM	oEVs were internalized in embryo, increasing blastocyst rate and embryo quality	[73]
2b	oviduct	*bos taurus*	ultracentrifugation	Embryo	NTA, TEM	oEVs increased embryo quality and altered expression of SNRPN	[74]
3	follicular fluid	*bos taurus*	ultracentrifugation	Embryo	tRPS, TEM	FF isolated EVs caused transcription and epigenetic alteration in embryos	[75]
4	embryo	*sus scrofa*	ultracentrifugation and precipitation	endometrial epithelial cells	TEM	evidence on embryo endometrial cross-talk mediated by EVs. EVs released by trophectoderm-cells increase the expression of miRNAs in maternal endothelial cells related to angiogenesis signaling	[76]
4	uterine flushings (UFs) from pregnant ewes	*ovis aries*	polymer precipitation	endometrial epithelial cells	TEM, NTA	Conceptus-derived EVs induce the expression of Interferon-Stimulated Genes ISG in bovine EECs culture	[77]
5	cytotrophoblast cell-derived exosome	*homo sapiens*	ultracentrifugation	extravillous trophoblasts (EVT)	TEM	Exosomes regulate intercellular communication between placental cells and EVT cell invasion in an oxygen-dependent manner	[78]
6a	Endometrial tissue and uterine fluid	*homo sapiens*	ultracentrifugation	Embryo	tRPS, FC	Endometrial derived EVs contain specific miRNA that may contribute to the endometrial-embryo cross talk and embryo implantation	[1]
6a	endometrial epithelial cell	*homo sapiens/mus musculus*	ultracentrifugation	Embryo	TEM	Endometrial derived EVs transport miRNAs influencing embryo transcriptomic for genes related to embryonic adhesion phenomenon	[8]
6a	Uterine Fluid	*bos taurus*	polymer precipitation	Embryo	TEM, NTA	EVs from uterine fluid regulate bovine conceptus implantation	[79]
6a	Uterine Fluid	*bos taurus*	polymer precipitation	Embryo	EM	EVs from uterine fluid of cows with endometritis impact blastocyst development	[80]
6b	Uterine Fluid	*gallus gallus*	ultracentrifugation	Sperm	TEM	Uterine fluid EV contain proteins that may play an essential role in the preservation of sperm functions	[81]

**Table 3 biomolecules-10-01510-t003:** Studies reporting the use of molecular cargos of EVs as biomarkers in reproductive medicine. Studies were categorized by different pathological conditions describing the source and method of EV isolation and the main results obtained.

Biomarkers for:	EVs Isolated from	Species	Isolation Method	Main Results	References
female reproductive cancer	serum	*homo sapiens*	polymer precipitation	EV miRNAs increase in the serum of epithelial ovarian cancer patients	[123]
female fertility	follicular fluid	*homo sapiens*	ultracentrifugation	EV-miRNAs in follicular fluid are associated with urinary concentrations of phenols and phthalate metabolites	[124]
female fertility	medium of blastocysts and endometrial cell co-cultures	*homo sapiens*	polymer precipitation	EV-bound secreted miRNAs are altered in co-culture experiments with blastocysts and endometrial cells isolated from patients diagnosed with AMA or endometriosis	[125]
embryo quality	medium of embryo cultures	*homo sapiens*	no isolation	DNA content in EVs isolated from embryo culture is linked to successful implantation	[126]
placenta quality	primary cytotrophoblasts and serum	*homo sapiens*	polymer precipitation and ultracentrifugation	serum EVs from patients with preeclampsia showed alteration in syncytin content	[127]
placenta quality	plasma	*homo sapiens*	ultracentrifugation	microRNAs from plasma EVs are altered in preeclampsia	[128]
placenta quality	plasma	*homo sapiens*	polymer precipitation and ultracentrifugation	EVs from preeclampsia patients delivered antiangiogenic factors to endothelial cells	[102]
placenta quality	plasma	*homo sapiens*	Ultracentrifugation and size exclusion chromatography	Proteomic analysis of plasma EVs revealed protein alterations related to gestational diabetes mellitus	[129]
placenta quality	condition media of chorionic villi	*homo sapiens*	Ultracentrifugation	Gestational diabetes mellitus alters miRNA profile of EVs isolated from chorionic villi explant cultures	[130]
early abortion	serum	*bos taurus*	Ultracentrifugation	EVs from serum contain microRNAs related to embryonic mortality in cows	[131]

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
