# Peer review of "The Biological Function of Extracellular Vesicles during Fertilization, Early Embryo—Maternal Crosstalk and Their Involvement in Reproduction: Review and Overview"

_biomolecules, 2020, doi:10.3390/biom10111510_

Round 1

Reviewer 1 Report

The manuscript is much improved, however still there are minor issues to update. This relates to the depth and specificity of references and content to the field. If references can be updated this will significantly add to the review. Note we would suggest updating references to the last 2-3 years, as there are several areas in the field that are not covered

  • Line 43 – insert Extracellular vesicles (EVs) are membrane bound organelles which can convey information between cells through the transfer of functional protein and genetic information to alter phenotype and function of recipient cells (REF: J Clin Invest 126, 1152-1162, doi:10.1172/JCI81129 (2016). doi:10.1146/annurev-cellbio-101512-122326 (2014). PMID: 32029601,

Line 45 – this reference alone does not support this – suggest inclusion of several other KEY refs – e.g., PMID: 32402079, PloS one 8, e58502, doi:10.1371/journal.pone.0058502 (2013), PMID: 26395145, PMID: 29846695, Proteomics 19, e1800423, Biology of reproduction 94, 38, doi:10.1095/biolreprod.115.134890 (2016). doi:10.1002/pmic.201800423 (2019). PMID: 31825151 (reviewed PMID: 29390102 PMID: 26709898). Note several other KEY refs should be included here - expand

  • Line 82-83 – update references (last 2 years)
  • Line 84-88 – update references (exomere, https://doi.org/10.1016/j.celrep.2019.01.009, https://www.nature.com/articles/s41556-018-0040-4, and https://pubmed.ncbi.nlm.nih.gov/30833697/)
  • Line 146 – Incl ref 31 + PMID: 30457403
  • Line 159 – suggest inclusion of ‘surfaceome’ studies investigating specific surface profile of EVs – https://link.springer.com/article/10.1007/s00281-018-0682-0, https://www.nature.com/articles/s41467-019-11486-1; and several others in the past 2 years
  • Table 2 – inclusion of additional references in the field – for example, but not limited to PMID: 32712384, PMID: 32350983, PMID: 31196738, PMID: 29196267, PMID: 27317346, PMID: 23516492
  • Line 247-251 – has profiling studies shown LIF present in EVs?
  • Line 251 – addition of key reference and discussion - PMID: 32402079. Suggest inclusion of this reference also in Section 8
  • Line 561-570 – inclusion of key references

After these corrections have been updated, I am happy for acceptance

Author Response

REVIEWER 1

The Authors thanks the Reviewer for considerations and helpful suggestions. According to the comments and suggestions, we have carefully evaluated all critical points and the manuscript has been thoroughly revised. The Authors hope that now the manuscript is suitable for publication on “Biomolecules”.

The manuscript is much improved, however still there are minor issues to update. This relates to the depth and specificity of references and content to the field. If references can be updated this will significantly add to the review. Note we would suggest updating references to the last 2-3 years, as there are several areas in the field that are not covered

Line 43 – insert Extracellular vesicles (EVs) are membrane bound organelles which can convey information between cells through the transfer of functional protein and genetic information to alter phenotype and function of recipient cells (REF: J Clin Invest 126, 1152-1162, doi:10.1172/JCI81129 (2016). doi:10.1146/annurev-cellbio-101512-122326 (2014). PMID: 32029601,

  • Thank you very much for your suggestion. We have integrated the text as suggested and we have inserted the relative references.

Line 45 – this reference alone does not support this – suggest inclusion of several other KEY refs – e.g., PMID: 32402079, PloS one 8, e58502, doi:10.1371/journal.pone.0058502 (2013), PMID: 26395145, PMID: 29846695, Proteomics 19, e1800423, Biology of reproduction 94, 38, doi:10.1095/biolreprod.115.134890 (2016). doi:10.1002/pmic.201800423 (2019). PMID: 31825151 (reviewed PMID: 29390102 PMID: 26709898). Note several other KEY refs should be included here – expand

  • Thank you very much for your suggestion. We have included the references suggested to better support the sentence.

Line 82-83 – update references (last 2 years)

  • Thank you very much for your suggestion. We have added two references that reviewer last methods used for EVs isolation.
  • Konoshenko, M.Y.; Lekchnov, E.A.; Vlassov, A.V.; Laktionov, P.P. Isolation of Extracellular Vesicles: General Methodologies and Latest Trends.  Biomed Res Int. 2018, 2018, 8545347. doi: 10.1155/2018/8545347.
  • Sidhom, K.; Obi, P.O.; Saleem, A. A Review of Exosomal Isolation Methods: Is Size Exclusion Chromatography the Best Option? Int J Mol Sci. 2020, 21, 6466. doi: 10.3390/ijms21186466.

Line 84-88 – update references (exomere, https://doi.org/10.1016/j.celrep.2019.01.009, https://www.nature.com/articles/s41556-018-0040-4, and https://pubmed.ncbi.nlm.nih.gov/30833697/)

- Thank you very much for your suggestion. We have updated the text with  the suggested references.

Line 146 – Incl ref 31 + PMID: 30457403

- Thank you very much for your suggestion. We have added the suggested references to the text.

Line 159 – suggest inclusion of ‘surfaceome’ studies investigating specific surface profile of EVs – https://link.springer.com/article/10.1007/s00281-018-0682-0, https://www.nature.com/articles/s41467-019-11486-1; and several others in the past 2 years

- Thank you very much for your suggestion. We have added the suggested references to the text.

Table 2 – inclusion of additional references in the field – for example, but not limited to PMID: 32712384, PMID: 32350983, PMID: 31196738, PMID: 29196267, PMID: 27317346, PMID: 23516492

  • We have added the references in the table requested except for:

PMID: 32712384,  because it is referring to a review whereas Table II reported only specific studies

PMID: 27317346, because it illustrated different methods for EVs isolation from uterine fluids without reporting specific targeting.

Line 247-251 – has profiling studies shown LIF present in EVs?

- Yes, we have found it and we have added the relative reference

Line 251 – addition of key reference and discussion - PMID: 32402079. Suggest inclusion of this reference also in Section 8

- Thank you very much for your suggestion. We have added the suggested references to the text.

Line 561-570 – inclusion of key references

  • Thank you very much for your suggestion, but we prefer to not add any references in the conclusions.

We feel that we have addressed all the queries raised by the referees and hope that the paper is now acceptable for publication in Biomolecules.

We thank you in advance for your time and consideration.

On behalf of all authors best regards,

Anna Lange-Consiglio and Emanuele Capra

Reviewer 2 Report

This is well-written and interesting review presenting EVs also as potential non-invasive biomarkers of specific pathologic condition and as a tool of innovative therapies in reproductive medicine.

I just have some minor corrections:

L71 ‘exosomes’ instead of exoxomes

L83 redundant comma in the bracket

L111 should be taken into consideration…  italics?

L220-221 symbol α is missing…  α6ß1 …α7ß1..  α4ß1

L318 … Cuman et al. (2015)… number of reference is missing [95]

L359 …Table 3 instead of Table 2

L367 …’dependent on’

L448 [138]…missing bracket

In vitro, in vivo…inconsistent font within the whole manuscript

Author Response

REVIEWER 2

The Authors thanks the Reviewer for considerations and helpful suggestions. According to the comments and suggestions, we have carefully evaluated all critical points and the manuscript has been thoroughly revised. The Authors hope that now the manuscript is suitable for publication on “Biomolecules”.

This is well-written and interesting review presenting EVs also as potential non-invasive biomarkers of specific pathologic condition and as a tool of innovative therapies in reproductive medicine.

I just have some minor corrections:

L71 ‘exosomes’ instead of exoxomes

  • We thank the referee for the suggestion and we have corrected the word

L83 redundant comma in the bracket

  • We thank the referee for the suggestion and we have corrected

L111 should be taken into consideration…  italics?

  • We thank the referee for the suggestion, and we have corrected

L220-221 symbol α is missing…  α6ß1 …α7ß1..  α4ß1

  • We thank the referee for the suggestion and we have corrected

L318 … Cuman et al. (2015)… number of reference is missing [95]

  • We thank the referee for the suggestion, the number of reference has been reported at the end of the sentence. We also have added the reference just after Cuman et al. (2015) in the line.

L359 …Table 3 instead of Table 2

  • We thank the referee for the suggestion and we have corrected

L367 …’dependent on’

  • We thank the referee for the suggestion and we have corrected

L448 [138]…missing bracket

  • We thank the referee for the suggestion and we have corrected

In vitro, in vivo…inconsistent font within the whole manuscript

  • We thank the referee for the suggestion and In vitro, in vitro…have been uniformed in the text.

We feel that we have addressed all of the queries raised by the referees and hope that the paper is now acceptable for publication in Biomolecules.

We thank you in advance for your time and consideration.

On behalf of all authors best regards,

Anna Lange-Consiglio and Emanuele Capra

This manuscript is a resubmission of an earlier submission. The following is a list of the peer review reports and author responses from that submission.

Round 1

Reviewer 1 Report

In this review, the authors summarized the function of EVs in embryo-maternal interaction during early pregnancy in mammals, which is an attracting topic for reproductive scientists. At first, they introduced the biogenesis of EVs, general methods used for isolation and characterization of EVs, as well as cargo content of EVs in general. As there are already specialized review articles focusing on each of these aspects in depth, it would be benefit if the authors could really emphasize and give hints on reproduction-related EVs, e.g. methods used for follicular fluid/oviductal fluid preparation, work with embryo derived EVs which are limited by their amount, common or special markers of EVs originate from reproductive tract among different species, and etc.

Later, authors reviewed recent studies and summarized the molecular cargoes, and function of EVs in reproduction and embryo‐maternal cross‐talk in human and animals, and proposed their clinical applications. According to the `MISEV2018` stated by the ISEV society, to prove a detected function is EV-borne, researchers should demonstrate the specific association of the activity with EVs rather than with co-isolated components or other soluble mediators. Considering that many studies in the reproduction field, especially during early years, used isolation methods which also co-isolated other non-EV components, it is necessary to mention this in the review, and special caution should be given when interpreting the findings of studies.

Major comments:

  1. Authors inconsistently used EVs, Ex, or exosomes throughout the manuscript, which is confusing for readers. According to the `MISEV2018`, ``assigning an EV to a particular biogenesis pathway remains difficult… unless authors can establish specific markers of subcellular origin``. Use of the generic term “extracellular vesicle” (EV) is suggested.
  2. Grammar: in many parts of the text, inconsistent use of present tense and past tense.
  3. Major issues in figure 2:
  4. From the figure, it seems that spEVs were only present in ampulla region of oviduct, then transport to the uterus? In the paper by Bai et al 2018, authors actually isolated EVs from the boar semen, and then treated porcine endometrial epithelial cells (EECs) in vitro.
  5. What is the difference between 2a and 2b?
  6. I assume that authors use the human female reproductive tract as example, since the situation is different among species.
  7. Table 1:
  8. Abbreviation EEC is not explained.
  9. ``PMCA4 sperm up‐take enhance sperm motility and fertility``. I think the results is somehow overstated, as the author did not perform sperm motility analysis in the paper.
  10. Bathala et al. was published in 2018?
  11. Study by Lopera‐Vasquez et al. was in 2017. They reported only isthmus oEVs increased embryo quality, only SNRPN was significantly altered. TEM was also used for EV characterization.
  12. I think there are several misinterpretations of the study by Nakamura et al., for example: the EV source, species, finding…
  13. Line 186: in reference 65, the author performed actually cross-species study.
  14. Line 245-253: I think these studies cannot really prove that´s EV related, therefore not much relevant to the topic.
  15. In section 7.1. Biomarkers for female reproductive cancer., author
  16. In section 8, can authors talk more about the therapeutic action of EVs in the reproduction field?

Minor comments:

Line 163: add space between text and reference.

Line 266: ``in vitro studies``. One reference is provided.

Line 271: EVs d?

Line 280-281: the sentence is hard to understand.

Table 2: Jayabalan et al. was published in 2019. Why author say it is biomarker for placenta quality?

Line 333: change cell to cells.

Line 379: tolerance to the fetus?

Line 392-393: ``Indeed, variation in oxygen tension and glucose can increase exosome release and affect both content and bioactivity of cell targets [69, 128]``. The statement is not clear.

Line 395: add space after reference.

Line 426: ``The EVs … can be loaded with drugs (in cells or in EVs)``. The sentence is not clear.

Line 429: Reference is missing.

Reviewer 2 Report

This review aims to summarise the novelty and clinical potential of EVs in the context of embryo-maternal crosstalk.  The review covers an important and emerging aspect of the field – namely the contribution of EVs to this area of reproductive biology. However, the review misses a lot of key areas in the field, key developments and importantly the challenges that remain in this field.

1 - Title does not fit review. Review highlights a lot on a broad range of topics including reproductive pathology and fertilisation, but should have more focus on the potential of embryo- and maternal-EVs since the review is on “the biological function of EVs in embryo-maternal communication in the early stages of mammalian conception”. As such, the review could be refocused to address the actual title. There are >10 key studies which are missing from the review in this specific area – i.e., maternal derived EVs (cargo, function, purification, hormonal priming and regulation), embryo/trophectoderm-derived EVs (cargo, function, transfer), and uterine and IVF media and the contribution of EVs in these areas. For example EVs in the context of embryo-maternal communication: I.e., Giacomini for human embryo-derived EVs, Greening for human endometrial-derived EVs, Evans for human trophectoderm for example. List of studies that have demonstrated either the secretion, transfer, and/or functional influence of EVs at this interface, including the implantation and maybe early placentation stages.

2 - Some background on the biology required. What is happening at the embryo-maternal interface – why is the emphasis of the title placed here? Why is communication important here? How was communication achieved before EVs were introduced? Inclusion of soluble and other types of communication is also missing. I.e., LIF, interleukins, HB-EGF, integrins, cell surface receptors and other factors that mediate embryo-maternal communication in early conception. How and when do follicular fluid, oviductal fluid, and uterine fluid come into play, and why are they important? What is the concentration of these components (i.e., EVs, types of EVs, and soluble factors for example).

3 - The authors have outlined various isolation and characterisation strategies of EVs, but as this paper is EV-focused, comparisons (pros and cons) of each technique should be included (so that the audience get a general idea of the techniques in the field and how they can be modified according to sample type/quantity/downstream studies). Each of these techniques and approaches are context dependent – therefore this should be raised and discussed in the context of the EV field (i.e., guidelines presented by international EV body, ISEV). Importantly, the methods that have been used in studies that have explored the role of EVs in reproductive biology (including, but not limited to ultracentrifugation, polymer precipitation could be presented in a table, noting key references and +/- for approach/technique).

4 - More emphasis on the importance of EV characterisation is required. Especially since authors have listed two major subtypes with an emphasis on different biogenesis and potential functions. What are the small EV/exosome-enriched markers and other techniques used to characterise exosomes as per ISEV guidelines? Further considerations and limitations need to be addressed regarding the characterisation of shed microvesicles/large EVs. What other techniques have been utilised to characterise these EVs? There is a lot more to be added in molecular characterisation.

5 – There is no comment on how EVs can alter target cells and therefore alter embryo-maternal crosstalk. This requires further discussion and key papers.

6 – The review covers the role of EVs in this crosstalk but does not include further discussion about other cell types and communication in this setting. For example, immune cell, stromal cell contributions? What are the challenges in the field moving forward and how will these be addressed? For example, dose and concentration of EVs, level of purification, limitations with production, as examples.

7 - Lacking significance when describing studies. While this review summarises the findings of key studies, a concluding statement stating how this contributes to the scope of embryo-maternal communication should be included. The review presents an overview of studies but not specifically how the study contributes to key areas of the field (i.e., minimal perspective). I.e., 185, trophoblast released EVs containing xyz to modulate expression of genes in the endometrium, what has changed? Why is it important that the trophoblast can do this?

8 - References required to back up several statements throughout the review. I.e., 108, 116, 163, 171, 184, etc.

Round 2

Reviewer 1 Report

In the revised version, the authors have addressed most of the main issues. As mentioned in the 1st round of revision, I still suggest to remind readers about the fact that ‘many studies in the reproduction field, especially during early years, used low-specificity isolation methods which also co-isolated other non-EV components, and therefore caution should be given when interpreting the findings of studies (i.e. the detected function is specifically EV-borne`).

Point 1.  Table 1, authors only listed several common isolation methods, but techniques such as microfluidic technologies, nanoscale flow cytometry are missing. Please specify.

Point 2.  In table 1, introduction on density gradient method is not precise. The method is not limited to sucrose, other chemicals for example iohexol, or iodixanol are also often applied. Samples could be loaded either bottom (float up) or on top (move downwards).

Point 3. Table 1, filtration, please correct ‘be lost to analysis’.

Point 4. Table 1, precipitation method is not limited to PEG. In many literatures, EVs have been precipitated with organic solvents, sodium acetate, or protamine.

Point 5. Table 1, I think in comparison to many other methods, processing time for size-exclusion chromatography is much faster.

Point 6. In part 3 methods for characterization of EVs: authors just mentioned a few techniques, while routine methods such as cryo-EM, Tunable Resistive Pulse Sensing (TRPS), dedicated flow cytometry and etc. are missing in this part.

Point 7. Line 357-358, I disagree with this statement. I think the dialogue between the embryo and the female reproductive tract already begins in the oviduct.  Typo error ‘femal’.

Point 8. Figure 2 interaction of oviductal tract EVs (oEVs) with sperm (2a) and embryo (2b). I suggest authors to remove the arrows, because the interaction with EVs constantly exists inside the whole tube, no matter the ampulla or isthmus region. The arrow indicates a clear direction, is somehow misleading.

Reviewer 2 Report

The revised version of this review has addressed several of the concerns raised. However, the article is still lakcing on the depth of analysis and reference to critical papers. Specifically, the reivew does not interpret or explain WHY the study is important, the approach or models used, and the contribution to the FIELD. These inclusions are required throughout the manuscript. The current version still reads like an overview, and not a specific review which is a key contributor to the field. For example, the role of EVs in biofluids you have addressed but in the context of the biology why is this of interest remains absent from the review. LIF is mentioned to be important - but how? what are downstream effectors of LIF? what is the context of this regulation in biology. Further, the dose of EVs, the physiological concentration of EVs, the challenges in the field of EVs remains very limited in the review. These points must be addressed in detail.

We feel that this minor update to the review does not specifically address many of the concerns raised initially. As such, we do not feel that the current version is acceptable for publication in its current form
